# PASSAGE RANKING WITH WEAK SUPERVISION

**Peng Xu   Xiaofei Ma   Ramesh Nallapati   Bing Xiang**
AWS AI
{pengx,xiaofeim,rnallapa,bxiang}@amazon.com

## ABSTRACT

In this paper, we propose a *weak supervision* framework for neural ranking tasks based on the data programming paradigm (Ratner et al., 2016), which enables us to leverage multiple weak supervision signals from different sources. Empirically, we consider two sources of weak supervision signals, unsupervised ranking functions and semantic feature similarities. We train a BERT-based passage-ranking model (which achieves new state-of-the-art performances on two benchmark datasets with full supervision) in our weak supervision framework. Without using ground-truth training labels, BERT-PR models outperform BM25 baseline by a large margin on all three datasets and even beat the previous state-of-the-art results with full supervision on two of the datasets.

## 1 INTRODUCTION

Recent advances in deep learning have allowed promising improvement in developing various state-of-the-art neural ranking models in the information retrieval (IR) community (Guo et al., 2017; Xiong et al., 2017; Mitra et al., 2017; Hui et al., 2018; Nogueira & Cho, 2019). Similar achievement has been seen in the reading comprehension (RC) community using neural passage ranking (PR) models for answer selection tasks (Yu et al., 2014; Tan et al., 2015; Yang et al., 2018; Lai et al., 2018). Most of these neural ranking models, however, require a large amount of training data. As such, we have seen the progress of deep neural ranking models is coming along with the development of several large-scale datasets in both IR and RC communities, e.g. (Bajaj et al., 2016; Feng et al., 2016; Dietz et al., 2017; Cohen et al., 2018). Admittedly, creating hand-labeled ranking datasets is very expensive in both human labor and time.

To overcome this issue, one strategy is to utilize weak supervision to replace human annotators. Usually we can cheaply obtain large amount of low-quality labels from various sources, such as prior knowledge, domain expertise, human heuristics or even pretrained models. The idea of weak supervision is to extract signals from the noisy labels to train our model. Dehghani et al. (2017) first applied weak supervision technique to train deep neural ranking models. They show that the neural ranking models trained on labels solely generated from BM25 scores can remarkably outperform the BM25 baseline in IR tasks. Macavaney et al. (2017) further investigated this approach by using external news corpus for training.

In this work, we focus on the setting where queries and their associated candidate passages are given but no relevance judgment is available. Instead of solely relying on the labels from single source (BM25 score), we propose to leverage the weak supervision signals from diverse sources. Ratner et al. (2016) proposed a general data programming framework to create data and train models in a weakly supervised manner. To tailor to the ranking tasks, instead of generating a ranked list of passages for each query, we generate binary labels for each query-passage pair. In our neural ranking models, we focus on BERT-based ranking model (Devlin et al., 2018) (architecture shown in Fig. 1), which achieves new state-of-the-art performance on two public benchmark datasets with full supervision. The contributions of this work are in two fold: (a) we propose a simple data programming framework for ranking tasks; (b) we train a BERT ranking model using our framework, by considering two simple sources of weak supervision signals, unsupervised ranking methods (BM25 and TF-IDF scores) and unsupervised semantic feature representation, we show our model outperforms BM25 baseline by a large margin (around $20\%$ relative improvement in top-1 accuracy on average) and the previous state-of-the-art performance (around $10\%$ relative improvement in top-1 accuracy on average) on three datasets without using ground-truth training labels.

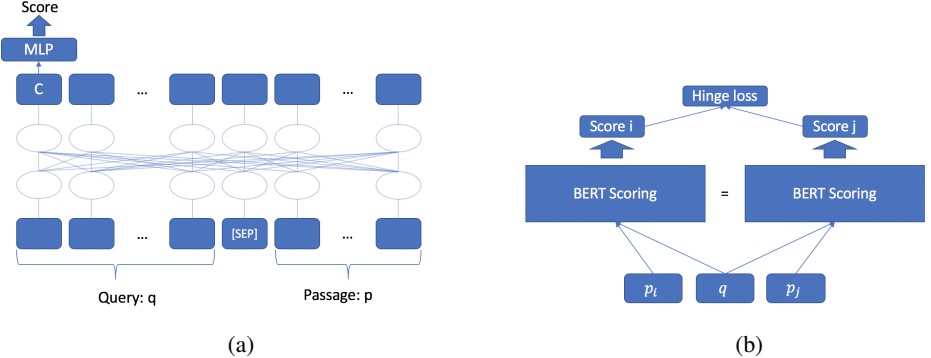

(a)                                                    (b)

Figure 1: BERT-PR model architecture. (a) depicts the architecture of BERT scoring model. (b) depicts the forward computation pipeline of training our ranking model.

## 2 APPROACH

In this section, we will describe in detail how we train a neural ranking model using weak supervision. We begin with introducing our BERT-PR model in Section 2.1. Then in Section 2.2 we will describe the weakly supervised training pipeline.

### 2.1 PASSAGE-RANKING WITH BERT

The goal of a ranking model is to estimate the (relative) relevance of a set of passages $\{p_i\}$ to a given query $q$. Here we apply BERT as our scoring model, which measures the relevance score of a candidate passage $p_i$ and the query $q$. Similar to the setup of sentence pair classification task in Devlin et al. (2018), we concatenate the query sentence and the candidate passage together as a single input of the BERT encoder. We take the final hidden state for the first token ([CLS] word embedding) of the input, and feed it into a two-layer feedforward neural network (with hidden units 100,10 in each layer and ReLU activation). The final output will be the relevance score between the input query and input passage. In the supervised setting, we assume we have the ground truth relevance label of the candidate passages for each query in the training set. To train our BERT ranking model, we use pairwise hinge loss. Specifically, for each triplet $\{q, p_i, p_j\}$, where $q$ is the query and $p_i, p_j$ are two candidate passages, the train loss of this instance is

$$\ell(q, p_i, p_j; \theta) = \max\{0, \epsilon - \text{sign}(\text{Pos}_{p_i} - \text{Pos}_{p_j})(\mathcal{S}(q, p_i; \theta) - \mathcal{S}(q, p_j; \theta)\} \tag{1}$$

where $\text{Pos}(p)$ indicates the ground truth ranking position of candidate passage $p$, $\mathcal{S}$ is our BERT scoring model as described and $\epsilon$ is the hyperparameter that determines the margin of hinge loss. Note that our BERT-PR is different from (Nogueira & Cho, 2019) in two aspects: (1) Nogueira & Cho (2019) uses cross-entropy loss instead of ranking hinge loss for training; (2) Nogueira & Cho (2019) does not have an MLP module. The details of our BERT-PR model is illustrated in Fig. 1.

### 2.2 WEAK SUPERVISION FOR PR

Now we present the weak supervision training pipeline for PR tasks. The main idea follows the paradigm in Ratner et al. (2016), which contains three major steps: (a) defining labeling functions that can generate noisy labels on the datasets (without true labels), (b) aggregating all the noisy labels to generate potentially more accurate labels as well as more coverage, (c) using the aggregated label to train a supervised model.

**Labeling Functions**   Ideally, we require a ranked list for each query in our training set for supervised training. However, obtaining accurate ranking labels for all sets of documents is very difficult. Instead, we reduce the task to a simpler problem, labeling whether the a candidate passage is strongly related to the query. With the binary label on the question-passage pair, it is easy to generate triplet training instance by doing positive and negative sampling. Formally, the labeling function is defined as $\lambda : \mathcal{Q} \times \mathcal{P} \rightarrow \{1, -1, 0\}$, i.e. for each query-passage pair, we would like to label it as positive, negative or neutral (undetermined). We first define some score function to measure the similarity of query-passage pair. Considering that the similarity scores across different queries may not be comparable, we categorize passages based on each individual query. Specifically, for each query, we rank the candidate passages based on the similarity scores and we take the top-1 passages as positive ones, the bottom half as negative ones, and label the rest in this list as neutral. With this

schema, we obtain $\{(q_i, p_j^{(i)}), y_{ij}\}$ for every query-passage pair $(q_i, p_j^{(i)})$ with $y_{ij} \in \{1, -1, 0\}$. In this work, we apply 4 scoring functions: (1) *BM25 score*, (2) *TF-IDF score*, (3) *cosine similarity of universal embedding representation* (Cer et al., 2018) and (4) *cosine similarity of the last hidden layer activation of pretrained BERT model* (Devlin et al., 2018).

**Label Aggregation**  This step is to aggregate all the weak supervision signals from all the labeling functions. Each label function may produce low quality labels. The step can be considered as an ensemble step to improve the quality of labels. We consider two simple strategies. The first one is through majority voting, i.e., we assign the final label based on the majority agreement, with the majority fraction as the confidence score. The second strategy is to learn a simple generative model based on the assumption that the labeling functions are conditionally independent given the true label. We apply the same parameterization as proposed in Ratner et al. (2016); see details in Appendix A. We predict the final label based on the learned simple generative model.

**Supervised Training**  After label aggregation, we have a collection of query-passage pairs where each is associated with a binary label and confidence score, i.e. $\{q_i, p_j^{(i)}, y'_{ij}, s_{ij}\}$ where $y'_{ij} \in \{-1, 1\}$ and $s_{ij} \in [0, 1]$. In order to do supervised training, we can generate the triplet training instances by combining positive and negative pairs that share the same query through uniform sampling. For confidence score of the triplet, we simply take the geometric mean of confidence scores of original two pairs. Then we train our supervised model based on these labels.

## 3 EXPERIMENT AND RESULTS

We apply our approaches on three passage-ranking datasets, `WikipassageQA` (Cohen et al., 2018), `InsuranceQA_v2` (Feng et al., 2016), and `MS-MARCO` (Bajaj et al., 2016). In all these datasets, the groundtruth labels are binary, indicating whether the passage is relevant to the question. Table 1 shows the basic statistics of these datasets. In our weak supervision settings, we do not use any ground-truth labels or rank information of the datasets.

Table 1: Statistics of datasets used in the experiments. The number in parenthesis is the average number of passages associated with each question.

| Dataset | Train: #Q (#P/Q) | Val: #Q (#P/Q) | Test: #Q (#P/Q) |
|---|---|---|---|
| WikipassageQA | 3,332 (58.3) | 417 (62.9) | 416 (57.6) |
| InsuranceQA_v2 | 12,889 (500) | 2,000 (500) | 2,000 (500) |
| MS-MARCO | 398,791 (1,000) | 6,980 (1,000) | - |

**Training configuration**  In all the experiments, we use pretrained BERT *base* model from Devlin et al. (2018). For `WikipassageQA` dataset, we set the maximum sequence length to be 200 in BERT and batch size 64. For `InsuranceQA_v2`, we set the maximum sequence length to be 100 in BERT and batch size 128. For `MS-MARCO`, we set the maximum sequence length to be 70 in BERT and batch size 256. For all the training, we sweep over $\{1e-5, 2e-5, 3e-5\}$ for learning rate and the maximum number of training steps is 10,000. We use a learning rate warmup ratio of $0.1$.

### 3.1 QUALITY OF PSEUDO LABELS

As we described in Section 2, we define four labeling functions. We adopt the retrieval component in DrQA(Chen et al., 2017) for the implementation of BM25 and TF-IDF scoring functions. We calculate cosine-similarity of BERT features and universal sentence embedding. To measure the quality of our labeling functions, we apply these labeling function on the training sets and compare our pseudo labels with the ground truth labels. Note that in ranking datasets, positive and negative pairs are highly imbalanced. So here we use precision and recall at 1 (P@1, R@1), and AUC to measure the quality of pseudo labels. The results are shown in Table 2. We learn the simple generative model (GM) over labeling functions to estimate the true label. Also we show the result of majority voting strategy. The quality of aggregated labels is shown in the bottom rows of Table 2.

### 3.2 PASSAGE RANKING PERFORMANCES

After aggregating the results of labeling functions, we now train our BERT-PR model. We compare the final performances of different models with different supervision signals along with the unsu-

Table 2: Quality of labeling functions. BM25: BM25 scores for each pair. TF-IDF: TF-IDF similarity scores for each pair. BERT: cosine similarity scores over the feature extracted from the last layer of hidden state of pretrained BERT base model. Universal: cosine similarity scores over features from universal sentence embedding. MV: labels after majority voting. GM: predicted labels of learned generative model.

| pseudo label | WikipassageQA | | | InsuranceQA_v2 | | | MS-MARCO | | |
|---|---|---|---|---|---|---|---|---|---|
| | P@1 | R@1 | AUC | P@1 | R@1 | AUC | P@1 | R@1 | AUC |
| BM25 | 48.25 | 29.09 | 77.25 | 22.48 | 19.11 | 85.38 | 31.30 | 30.02 | 81.21 |
| TF-IDF | 43.39 | 26.01 | 75.90 | 17.51 | 14.88 | 84.47 | 26.53 | 25.44 | 81.86 |
| BERT | 31.36 | 18.80 | 70.98 | 03.76 | 03.19 | 66.32 | 10.50 | 10.06 | 63.79 |
| Universal | 40.01 | 23.98 | 80.37 | 10.55 | 08.97 | 83.80 | 23.59 | 22.62 | 79.44 |
| MV | 48.74 | 29.21 | **83.54** | 21.72 | 18.46 | 85.44 | 40.93 | 39.24 | 89.08 |
| GM | 50.24 | 30.11 | 82.60 | 22.46 | 19.01 | **85.76** | 32.64 | 31.29 | **89.35** |

Table 3: PR performance on different datasets. SOTA: Cohen et al. (2018) for WikipassageQA, Rücklé & Gurevych (2017) for InsuranceQA_v2, Nogueira & Cho (2019) for MS-MARCO. The best numbers achieved by weak supervision models are in bold. * indicates the new SOTA performance in full supervision.

| method | WikipassageQA (test) | | | | InsuranceQA_v2 (test) | | | | MS-MARCO (validation) | | | |
|---|---|---|---|---|---|---|---|---|---|---|---|---|
| | MAP | MRR | P@1 | P@5 | MAP | MRR | P@1 | P@5 | MAP | MRR | P@1 | P@5 |
| BM25 (baseline) | 53.73 | 62.58 | 48.53 | 19.47 | 28.60 | 32.76 | 23.90 | 09.68 | 16.27 | 16.48 | 8.58 | 4.94 |
| full supervision | | | | | | | | | | | | |
| SOTA | 56.08 | 67.92 | - | 20.83 | - | 36.9 | - | - | - | 34.7[1] | - | - |
| BERT-PR | 73.55* | 80.87* | 70.19* | 27.35* | 45.32* | 49.59* | 40.05* | 15.19* | 34.53 | 35.00 | 23.02 | 10.10 |
| weak supervision with BERT-PR | | | | | | | | | | | | |
| BM25 | 57.25 | 65.47 | 52.16 | 21.11 | 32.25 | 36.29 | 26.50 | 11.08 | 18.65 | 18.97 | 9.86 | 5.77 |
| MV | **62.58** | **69.89** | **56.49** | 23.46 | 29.66 | 33.73 | 23.25 | 10.40 | 20.37 | 20.64 | **10.89** | 6.29 |
| MV (noise) | 61.62 | 69.31 | 54.67 | 23.58 | 30.03 | 34.09 | 24.10 | 10.35 | 20.12 | 20.27 | 10.16 | 6.30 |
| GM | 59.67 | 67.38 | 56.25 | 22.60 | **34.16** | **38.46** | **28.75** | **11.59** | 20.36 | 20.64 | 10.87 | 6.26 |
| GM (noise) | 59.04 | 66.98 | 52.16 | 22.12 | 33.18 | 37.31 | 27.20 | 11.52 | **20.44** | **20.70** | 10.86 | **6.33** |

pervised BM25 baseline. We use mean average precision (MAP), mean reciprocal rank (MRR), precision at 1 (P@1) and 5 (P@5) as our evaluation metrics. The results are shown in Table 3.

Note that through weak supervision solely on BM25 scores, BERT-PR already outperforms the unsupervised BM25 baseline, which is consistent with the results from Dehghani et al. (2017). In our training pipeline, using the simple generative model over the 4 labeling functions, BERT-PR trained on GM labels outperforms BM25 baselines as well as BERT-PR trained solely on BM25 scores. For example, in terms of P@1, BERT-PR trained on GM labels outperforms BERT-PR trained on BM25 by around 10% relatively on all three datasets. In the case of WikipassageQA and InsuranceQA datasets, our weak supervision models even beat the previous SOTA performances in the fully supervised settings, exhibiting the great potential of our weak supervision models in real applications. Also we report the results on supervised training on generated labels with confidence scores, as noise-aware training objective (See Eq. (4) in Appendix B), indicated by "noise" in the parenthesis. In our experiment, noise-aware training does not improve the performances significantly, probably because using geometric mean of scores of the pairs as the confidence scores of the triplets is not very good approximation of actual probability of generated labels. We leave this for future research.

## 4 CONCLUSIONS

In this work, we proposed a simple weak supervision pipeline for neural ranking models based on the data programming paradigm. In particular, we also proposed a new PR model based on BERT, which achieves new SOTA results. In our experiments on different datasets, our weakly supervised BERT-PR model outperforms the BM25 baseline by a large margin and remarkably, even beats the previous SOTA performances with full supervision on two datasets. Further research can be done on how to better aggregate pseudo ranking labels. In our pipeline we reduce the ranking labels into binary labels of relevance of query-passage pairs, which may result in loss of useful information. It would be interesting to design generative models on the ranking labels directly.

---

[1] The number listed in SOTA was reported on BERT base model for comparison. A better one was reported on BERT large model, which has MRR 36.5 (Nogueira & Cho, 2019).

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

## A    SIMPLE GENERATIVE MODEL FOR LABELING FUNCTIONS

In this section, we present the simple generate model on labeling functions and true labels as in Ratner et al. (2016). For completeness, here we restate the formulation as in Ratner et al. (2016). The basic model assumption is that given the true label, the labeling functions are conditionally independent. Formally, suppose $y \in \{-1, 1\}$ is the true label, $\lambda_1, \cdots, \lambda_k$ are the labels from $k$ labeling functions. The probabilistic graphic model is shown as in Fig. 2. Given the conditional independence assumption, we can parameterize the conditional distribution of the labeling function as follows:

$$Pr(\lambda_i|y) = \begin{cases} \beta_i \alpha_i, & \text{if } \lambda_i = y \\ \beta_i(1 - \alpha_i), & \text{if } \lambda_i = -y \\ 1 - \beta_i, & \text{if } \lambda_i = 0 \end{cases} \quad (2)$$

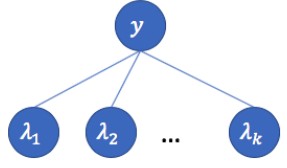

Figure 2: Simple Generative Model of labeling functions.

We also assume the prior of true label $y$ that $Pr(y = 1) = \gamma$ (In our experiment, for `WikipassageQA`, we set $\gamma = 0.01$, for `InsuranceQA_v2`, we set $\gamma = 0.002$, for `MS-MARCO`, we set $\gamma = 0.001$). Then we can find the optimal parameter $\alpha_i, \beta_i$ as maximizing the marginal likelihood of $(\lambda_1, \cdots, \lambda_k)$, i.e.

$$\alpha^*, \beta^* = \arg\max \frac{1}{N} \sum_{i=1}^{N} \log Pr(\lambda_1^{(i)}, \cdots, \lambda_k^{(i)}) = \arg\max \frac{1}{N} \sum_{i=1}^{N} \log \sum_{y_i} Pr(y^{(i)}) \prod_{j=1}^{k} Pr(\lambda_j^{(i)}|y^{(i)}) \quad (3)$$

where $N$ is the total number of data points. It is worth mentioning that given this formation, the model is not identifiable due to the symmetry of the model. A simple solution to remedy this issue is assuming $\alpha_i > 0.5$, meaning the most of labeling functions are doing right. With that, we can solve the problem Eq. (3) through projected gradient descent methods.

## B    NOISE-AWARE TRAINING OBJECTIVE

Ratner et al. (2016) introduces the noise-aware training objective to better incorporate the noise in the generated labels. The exact objective is as follows:

$$\mathcal{L}(\theta) := \frac{1}{N} \sum_{i=1}^{N} \ell(z_i; \theta) s_i, \quad (4)$$

where $s_i \in [0, 1]$ is the confidence score for data point $z_i$ being a correct training instance. For example, in our case of PR, $z_i$ is a triplet $(q, p_+, p_-)$ and $s_i$ is the confidence score of $p_+$ being more relevant to $q$ than $p_-$.

