# OpenReview forum: "Passage Ranking with Weak Supervision"
_ICLR.cc/2019/Workshop/LLD — LLD 2019_

### Official Review · AnonReviewer1 · 2019-04-13
**Good empirical contribution**

**Rating:** 4
**Confidence:** 2

**Review:**

The authors tackle the problem of passage ranking (i.e. given a query, rank the relevance of set of passages to this query), and propose using an interesting combination of two existing approaches: BERT, which has achieved state-of-the-art result on many similar NLP problems, and the weak supervision framework proposed by Ratner et al. (2016). The authors show that this combination obtains results that are better than the current fully supervised state-of-the-art.

Overall, although the different components of this system are not novel, this work seems to have a good contribution as an application paper since the results look good, and the topic is also very relevant to this workshop. However, my most major concern is the comparison with other similar approaches (in terms of methods and results). Specifically, there seems to be a related paper that is not properly discussed, nor fully compared with in terms of results (see my comments below).

Strengths:
- the problem is very relevant to this workshop.
- the results look good.
- the explanations are generally clear and easy to follow.


Major weaknesses:
- it sounds from the authors' description that the work of Nogueira & Cho (2019)  is very similar, and yet this paper doesn't discuss the similarities in enough detail. For instance "Nogueira & Cho (2019) does not have an MLP module" -- so what does it have instead?  Also, do they also do weakly supervised training?
- why are the results of Nogueira & Cho (2019) not reported in the table (except for one number in the footnote)?
- the citation for the most similar work is incomplete, it only says "Rodrigo Nogueira and Kyunghyun Cho. Passage Re-ranking with BERT. 2019.", with no information where to find it.
- it's unclear from this paper whether there are other weakly supervised approaches on these datasets, other than the traditional ranking scores the authors used as baseline. If the aren't, that should be specified. If there are, they should be compared and reported in the table too.

Minor issues:
- authors refer to BM25 scores without ever explaining what they are (e.g. "models trained on labels solely generated from BM25 scores"), which can be an issue for anyone who hasn't specifically worked in information retrieval.
- there are a few grammatical mistakes.
- why did the authors chose the hidden state of the CLS token as the embedding that is used as input to the MLP?
- what is s_ij in the "Supervised Training" paragraph of section 2?

---

### Decision · Program_Chairs · 2019-04-16
**Acceptance Decision**

Accept